# Secondary Treatment for Men with Localized Prostate Cancer: A Pooled Analysis of PRIAS and ERSPC-Rotterdam Data within the PIONEER Data Platform [note 1]

**DOI:** 10.3390/jpm12050751

**Published:** 2022-05-05

**Authors:** Katharina Beyer, Vera Straten, Sebastiaan Remmers, Steven MacLennan, Sara MacLennan, Giorgio Gandaglia, Peter-Paul M. Willemse, Ronald Herrera, Muhammad Imran Omar, Beth Russell, Johannes Huber, Markus Kreuz, Alex Asiimwe, Tom Abbott, Alberto Briganti, Mieke Van Hemelrijck, Monique J. Roobol

**Affiliations:** 1Translational and Oncology Research (TOUR), King’s College London, Faculty of Life Sciences and Medicine, London SE19RT, UK; beth.russell@kcl.ac.uk (B.R.); mieke.vanhemelrijck@kcl.ac.uk (M.V.H.); 2Department of Urology, Erasmus University Medical Center, 3015 Rotterdam, The Netherlands; vera.straten@outlook.com (V.S.); s.remmers@erasmusmc.nl (S.R.); m.roobol@erasmusmc.nl (M.J.R.); 3Academic Urology Unit, University of Aberdeen, Aberdeen AB24 3FX, UK; steven.maclennan@abdn.ac.uk (S.M.); s.maclennan@abdn.ac.uk (S.M.); m.i.omar@abdn.ac.uk (M.I.O.); 4Unit of Urology, Division of Oncology, URI, IRCCS Ospedale San Raffaele, 20132 Milan, Italy; giorgio.gandaglia@gmail.com (G.G.); briganti.alberto@hsr.it (A.B.); 5Department of Urology, Cancer Center, University Medical Center Utrecht, 3584 Utrecht, The Netherlands; ppmwillemse@hotmail.com; 6Integrated Evidence Generation-Data Science, Research & Analytics, Bayer AG, 13353 Berlin, Germany; ronald.herrera@bayer.com; 7Department of Urology, University of Technology, 01069 Dresden, Germany; johannes.huber@uniklinikum-dresden.de; 8Department of Diagnostics, Fraunhofer Institute for Cell Therapy and Immunology, 04103 Leipzig, Germany; markus.kreuz@izi-extern.fraunhofer.de; 9Department of Epidemiology, Bayer AG, 13353 Berlin, Germany; alex.asiimwe@bayer.com; 10Real World Evidence, Astellas Pharma, 2333 Leiden, The Netherlands; tom-abbott@astellas.com

**Keywords:** prostate cancer, active surveillance, treatment choice, patient decision-making, oncology, treatment selection

## Abstract

Introduction: Treatment choice for localized prostate cancer is complicated, as each treatment option comes with various pros and cons. It is well established that active surveillance (AS), may be ended with a change to curative treatment at the time of disease progression, but it is less clear whether secondary treatment after initial curative treatment is required. As part of the PIONEER project, we quantified the probabilities of treatment change. Methods: A cohort study based on PRIAS and ERSPC-Rotterdam data was conducted. Patients were followed up for 10 years or until the 31st of December 2017. The primary outcome was the incidence of treatment change following initial treatment (i.e., a change to curative treatment following AS or secondary treatment after initial RP/RT). Results: Over a period of 1 to 5 years after initial treatment, the cumulative incidence of treatment change ranged from 3.8% to 42.8% for AS, from 7.6% to 12.1% for radical prostatectomy (RP) and from no change to 5.3% for radiation therapy (RT). While the possibility of treatment change in AS is known, the numbers within a five-year period were substantial. For RP and RT, the rate of change to secondary treatment was lower, but still non-neglectable, with 5 (10)-year incidences up to 12% (20%) and 5% (16%), respectively. Conclusion: This is one of the first studies comparing the incidence of guideline-recommended treatment changes in men receiving different primary treatments (i.e., AS, RT, or RP) for localized prostate cancer (PCa).

## 1. Introduction

Prostate cancer (PCa) is the most frequently diagnosed cancer in men [1]. With more than 1.4 million diagnoses occurring in 2020 [1], the global burden of PCa is considerably high. The European Randomized Study of Screening for Prostate Cancer (ERSPC) was initiated to test whether early detection could help reduce mortality from PCa. This ongoing trial specifically aims to evaluate the effect of prostate-specific antigen (PSA) testing on PCa mortality and currently reports a 20% relative reduction in PCa mortality for men randomized to screening [2,3,4,5] Available online: (www.prias-project.org), (accessed on 1 February 2022). After 16 years of follow-up, the ERSPC reported a 41% higher (95% CI: 36–45%) PCa incidence in the screening arm compared to the control arm [2].

The curative therapeutic approach for men diagnosed with localized PCa is based on three main treatment options: radiation therapy (RT), such as external beam radiation; brachytherapy (internal radiation) and radiopharmaceuticals (wherein medicines which contain radiation are injected); and radical prostatectomy (RP) or active surveillance (AS), such as active monitoring instead of immediate treatment. Each of the treatment methods comes with various pros and cons, such as short- and long-term side effects with different impacts on patient health-related quality of life. Among the available therapeutic options, AS is characterized by an active monitoring of the disease followed by the use of curative-intent therapies in the case of progression, and is aimed at avoiding or postponing the long-term sequelae of RP and RT. In this context, it is well established that AS may be ended by a switch to curative treatment (i.e., RT or RP) at disease progression [6]. However, the frequency of adoption of secondary treatments after AS and their outcomes in early-stage PCa is still poorly assessed. Moreover, the frequency of adoption of secondary therapies in patients initially managed with RP or RT represents a main issue that has been poorly investigated.

In 2018, the Prostate Cancer Diagnosis and Treatment Enhancement Through the Power of Big Data in Europe (PIONEER) Consortium was launched by the Innovative Medicines Initiative 2 (IMI-2) as part of the Big Data for Better Outcomes Programme (BD4BO) [7,8]. The overarching goal of PIONEER is to provide high-quality evidence on PCa management by unlocking the potential of big data. The Rotterdam component of the above-mentioned ERSPC as well as the largest global web-based AS study, PRIAS [9], are parts of PIONEER [8]. Using these two data resources, we aimed to gain further insight into the frequency of (a) treatment switch after opting for AS, and (b) secondary treatments after RP or RT in men diagnosed with early-stage PCa.

A cohort study was conducted based on PRIAS and ERSPC-Rotterdam data included in the PIONEER data platform. The PRIAS study was initiated in 2006 and represents the daily clinical practice of men on AS. The 2006 inclusion criteria were adenocarcinoma of the prostate, PSA ≤ 10 ng/mL, PSA density <0.2 ng/mL/mL, clinical stage T1c-2, GS equal to or below 3 + 3, ≤2 positive biopsy cores, and fitness for curative treatment. Follow-up consisted of a 3-monthly PSA test and a 6-monthly digital rectal examination (DRE) during the first two years, followed by a 6-monthly PSA test and an annual DRE in the years after. Systematic biopsies based on prostate volume were scheduled at the 1st, 4th, 7th, and 10th year of follow-up and every five years afterwards. Curative treatment was recommended if a Gleason Score (GS) above 3 + 3 and more than two positive biopsy cores or clinical stage T3–4 were found.

The ERSPC-Rotterdam was initiated in 1993 and randomly assigned 42,376 men aged 55 to 70 years to a screening or control arm. The screening arm received four-yearly PSA-tests, and if PSA > 3.0 ng/mL was found in the DRE, transrectal ultrasonography and systematic sextant biopsies were recommended. If PCa was detected, screening was discontinued, and further evaluation and treatment were performed according to contemporary clinical practice. After treatment, PSA was regularly measured, and events of disease progression, treatment changes, and death were recorded by semi-annual chart reviews [2,3,4,5]. All methods were carried out in accordance with relevant guidelines and regulations (Declaration of Helsinki). All experimental protocols were approved by a named institutional and/or licensing committee.

## 2. Materials and Methods

### 2.1. Study Population

The first 500 PRIAS patients were selected for the AS group because of their long follow-up period (patients were followed up for 10 years or until 31 December 2017 (whichever came first)). To create comparable groups, inclusion criteria for patients from the ERSPC-Rotterdam were clinical stage T1c-2, GS equal to or below 3 + 3, PSA ≤ 10 ng/mL, and initial treatment with either RP or RT. To evaluate the possibility of a Will Rogers phenomenon [10], we conducted a sensitivity analysis in which we excluded patients from the ERSPC-Rotterdam who were diagnosed before the 2005 International Society of Urological Pathology (ISUP) consensus on Gleason Grading [11,12] (see Figure 1). 

### 2.2. Outcomes

In PIONEER, we developed a core outcome set (COS) for localized PCa and associated definitions [13,14]. This COS is applicable for effectiveness studies using randomized designs, observational designs, clinical audit, and big data studies.

The primary outcome was the incidence of treatment change following initial treatment (i.e., a change to curative treatment following AS or secondary treatment after initial RP/RT) in men with localized PCa. Treatment change was defined as biochemical recurrence (BCR) in the RP and RT groups. According to the American Urological Association (AUA), BCR after RP is defined as two consecutive PSA measurements (≥0.2 ng/mL) (21), a single PSA measurement ≥0.2 ng/mL followed by a therapy change, or a detectable PSA 3 months after RP. We defined treatment change in the RT group on the basis of the definition of BCR after RT in accordance with the American Society of Therapeutic Radiology and Oncology (ASTRO)—i.e., a PSA of 2 ng/mL or more above nadir. For AS, treatment change was defined as the discontinuation of AS based on protocol advice according to the PRIAS protocol.

### 2.3. Statistical Analysis

Descriptive statistics were performed to report baseline characteristics. For the initial treatment groups, follow-up was defined as the time between diagnosis and treatment change, death, last follow-up visit, or discontinuation of AS. For secondary treatment groups, follow-up was defined as the time between the discontinuation of AS and additional treatment, death, or last follow-up visit. Due to differences in initiation date between the ERSPC and PRIAS study, follow-up only included visits until 31 December 2017 or a maximum follow-up period of 10 years (whichever came first). Patients with only one follow-up visit in the RP and RT group were censored because they could never fit the definition of BCR. Cumulative incidences of treatment change were estimated for each treatment group, while death and discontinuation of AS not based on protocol advice were considered as competing risks [15]. All analyses were carried out using R version 3.5.1.

## 3. Results

### 3.1. Patient Characteristics

In total, 500 patients were included in the AS group, 557 in the RP group and 564 in the RT group (Table 1). Median age and PSA level at diagnosis for AS, RP, and RT were 66 (IQR, 61–70), 65 (IQR, 62–68), and 69 (IQR, 65–72) years, and 5.7 (IQR, 4.1–7.1), 4.6 (IQR, 3.4–6.4), and 5.0 (IQR, 3.7–7.1) ng/mL, respectively. PCa was found through PSA-based screening in 73%, 85%, and 77% of men treated with AS, RP, and RT, respectively.

### 3.2. AS Versus Initial RP and RT

Overall, the median follow-up of men without treatment change was 4.3 (IQR 1.0–7.2) years for men who opted for AS compared to 9.2 (IQR 6.7–9.8) years for men who opted for RP and 9.2 (IQR 6.1–10.0) years for men who opted for RT. Treatment change was observed in 245 men (49%) opting for AS, 94 men (17%) opting for RP, and 75 (13%) men opting for RT within median follow-up periods of 2.1 (IQR, 1.2–4.5), 1.6 (IQR, 0.5–5.5), and 5.8 (IQR, 3.7–7.8) years, respectively. After 1, 2, and 5 years, the cumulative incidences of treatment change were 3.8% (95% CI, 2.1–5.5), 25.8% (95% CI 21.8–29.8), and 42.8% (95% CI 38.2–47.4) for AS; 7.6% (95% CI 5.4–9.8), 9.6% (95% CI 7.1–12.1), and 12.1% (95% CI, 9.3–14.8) for RP; and 0.0% (95% CI, 0.0–0.0), 1.1% (95% CI 0.2–1.9), and 5.3% (95% CI 3.4–7.1) for RT, respectively, see Appendix A.

### 3.3. Secondary RP and RT after Discontinuation of AS

Secondary treatment follow-up data after the discontinuation of initial AS were available for 148 men. Of these 148 men, 73 switched to RP, 53 to RT, 16 to watchful waiting (WW), and in 6 patients, therapy was unknown. An additional treatment switch was observed in 26 patients, of whom 13 men were treated with RP, 7 with RT, 4 with WW, and 2 with an unknown therapy. Cumulative incidences of BCR following secondary treatment change after 1, 2, and 5 years follow-up for men having had RP after AS were 4.6% (95% CI, 0.0–9.7), 10.1% (95% CI 2.3–17.8), and 33.4% (95% CI, 16.5–50.3), and for men having had RT after AS, they were 1.9% (95% CI 0.0–5.6), 5.9 (95% CI 0.0–12.5), and 10.3% (95% CI 1.6–18.9). At 5 years follow-up, only 12 (16.4%) patients were still at risk in the RP group compared to 30 (56.6%) patients in the RT group, see Appendix A.

### 3.4. Sensitivity Analysis

We did not observe a Will Rogers phenomenon, as there were no improved clinical outcomes (e.g., lower cumulative incidences of BCR) (20) in the RP and RT groups compared to our primary analyses.

## 4. Discussion

Over a period of 1 to 5 years after initial treatment, the cumulative incidences of treatment change ranged from 3.8% to 42.8% for AS, from 7.6% to 12.1% for RP, and from no change to 5.3% for RT. These incidences are comparable to those of curative treatment after initial AS, with a 5-year cumulative incidence of 33.4% of those opting for RP and 10.3% for RT.

No formal RCT is available comparing protocol-based AS, RP, and RT. Nevertheless, the Prostate Testing for Cancer and Treatment (ProtecT) randomized trial applied, next to RT and RP, a less stringent AS protocol (active monitoring), which identified higher rates of disease progression in the active-monitoring-arm (21%) compared to the RP- (8%) or RT-arm (8%) after 10 years of follow-up [16]. These rates are lower than those identified in this study (27%, 17%, and 12% for AS, RP, and RT, respectively). This might be due to the different definitions used for disease progression. We linked treatment change to the definition of biochemical failure, while ProtecT was based on the definition on clinical failure (i.e., metastases, cT3–4, long-term androgen deprivation, ureteric obstruction, rectal fistula, or the need for urinary catheter) [17].

Several observational studies have studied progression after RP and RT. Kurbegovic et al., found a 10-year probability of BCR of 17.9% (95% CI 12.5–23.2%) in low-risk patients treated with RP, which is consistent with our 10-year probability of BCR for initial RP (20.2% (95% CI, 16.1–24.3) at 10 years) [17]. Kishan et al., found 5- and 10-year probabilities of BCR of 2.5% (95% CI 1.5–3.4) and 8.6% (95% CI 5.9%–11.2), respectively, in low-risk patients treated with stereotactic RT. These probabilities are lower than ours (5.4% (95% CI 3.5–7.3) at 5 years and 14.4% (95% CI 11.2–17.7) at 10 years, which may be explained by additional androgen deprivation and a higher dose given in stereotactic RT [18,19,20,21,22].

Kapadia et al., investigated time to BCR after RT and found no low-risk patients with a short interval (<1.5 years) to BCR [23]. Studies found a median time to BCR of 1.7–3.1 years after RP, and patients with early BCR had higher-risk disease [24,25]. However, besides the risk group, several other variables affect the likelihood of BCR, and differences in these other variables (e.g., positive surgical margins) may result in different intervals to BCR [25].

Godtman et al., evaluated BCR after RP in 132 patients who received initial treatment with AS and found BCR in 19% of the patients, which is consistent with our findings (13 out of 73) [26]. Other studies found overall BCR rates ranging from 7.7 to 50.4% after the discontinuation of AS [27,28,29]. Unfortunately, all these results were based on small patient cohorts and no comparison was made with initial curative treatment.

Our results indicate the importance of carefully informing PCa patients enrolled in an AS protocol or diagnosed with early-stage disease after PSA screening about what they can expect after their initial treatment. To achieve non-tokenistic shared decision making, the evidence-based presentation of outcomes including treatment change and side effects needs to be given to patients. Patients should be informed during consultation, but also through decision-making aids about the risk of treatment change, not just for AS, but also for RP and RT. This will help patients and HCPs set more realistic expectations.

Whilst our study benefits from large datasets with granular information, it is a limitation that different definitions for treatment change were used for different treatment modalities. For RP and RT, treatment change was defined on the basis of BCR, which is most often the driving force behind a change of treatment. The ASTRO definition for BCR after RT was chosen because of its correlation with clinical outcomes [30]. Our approach is pragmatic and makes best use of the available data. A limitation of the PIONEER COS is that it is difficult to retrospectively apply it to pre-existing datasets, as is the case in this study. Nonetheless, following our work on COS as part of PIONEER, we encourage all new studies to collect and report on the COS as a minimum or to justify why they are not collecting the COS, to make future critical summaries of the evidence base as informative as possible and relevant to all stakeholders, as well as to mitigate against selective outcome reporting, as is recommended by the COMET initiative [31].

In addition, the heterogeneity of our two study populations may be considered as a limitation. The RP and RT groups included patients diagnosed before the 2005 consensus on Gleason Grading [12], whereas the AS group comprised patients mostly diagnosed after this consensus. However, our sensitivity analysis did not indicate that differences in GS grading affected the results of our primary analysis. Finally, the treatment groups after the discontinuation of AS were small. At 5 years follow-up, only 12 (16.4%) patients remained at risk of BCR in the RP group, which complicates a reliable comparison with initial RP at that time point.

## 5. Conclusions

Treatment decision making in early PCa is challenging, since each treatment comes with different side effects which impact the QoL of patients and their families. This is one of the first studies comparing the incidence of guideline-recommended treatment changes in men receiving different primary treatments (i.e., AS, RT, or RP) for localized PCa. These data should become part of the evidence base that informs treatment decision making and be included in the shared decision-making process to avoid treatment regret caused by unrealistic expectations.

## Figures and Tables

**Figure 1 jpm-12-00751-f001:**
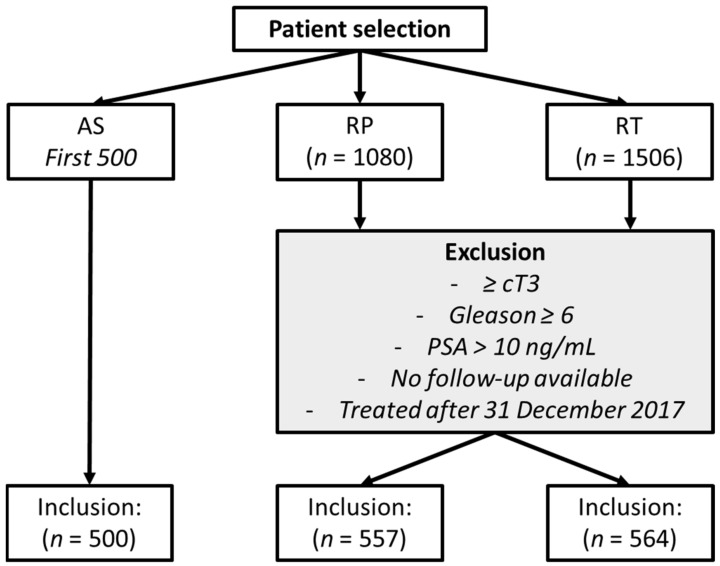
Patient flowchart. Active Surveillance (AS); prostatectomy (RP); radiation therapy (RT); prostate-specific antigen (PSA).

**Table 1 jpm-12-00751-t001:** Patient characteristics.

	AS			RP			RT		
	Overall	No Protocol Advice/Still on AS	Protocol Advice	Overall	No BCR	BCR	Overall	No BCR	BCR
*N* (no.)	500	255	245	557	463	94	564	489	75
Age (yr), median (IQR)	66 (61–70)	65 (60–70)	67 (62–71)	65 (62–68)	65 (62–68)	65 (61–69)	69 (65–72)	69 (65–72)	69 (66–73)
PSA (ng/mL), median (IQR)	5.7 (4.1–7.1)	6.0 (4.3–7.3)	5.4 (4.1–6.9)	4.6 (3.4–6.4)	4.6 (3.4–6.2)	5.1 (3.7–7.1)	5.0 (3.7–7.1)	4.9 (3.6–6.9)	6.4 (4.7–7.8)
PCa found by screening	367 (73%)	181 (71%)	186 (76%)	475 (85%)	396 (86%)	79 (84%)	434 (77%)	387 (79%)	47 (63%)
Follow-up until event * (yr), median (IQR)	2.9 (1.2–6.3)	4.3 (1.0–7.2)	2.1 (1.2–4.5)	9.0 (5.2–9.7)	9.2 (6.7–9.8)	1.6 (0.5–5.5)	8.7 (5.7–9.9)	9.2 (6.1–10.0)	5.8 (3.7–7.8)

IQR, interquartile range; PSA, prostate-specific antigen; BCP, biochemical progression; BCR, biochemical recurrence; AS, active surveillance; RP radical prostatectomy; RT, radiation therapy. * Follow-up until event is defined as time between diagnosis and BCP/BCR, discontinuation of AS, last follow-up visit, or death.

## Data Availability

Data are not available to other researchers since the data originate from patients providing routinely collected data.

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
