# Peer review of "Secondary Treatment for Men with Localized Prostate Cancer: A Pooled Analysis of PRIAS and ERSPC-Rotterdam Data within the PIONEER Data Platform†"

_jpm, 2022, doi:10.3390/jpm12050751_

Round 1

Reviewer 1 Report

By using PRIAS and ERSPC data, the authors try to compare the incidence of treatment changes in localized prostate cancer patients receiving different primary treatment, such as AS, RT or RP. Some issues were seen in the rationales and conclusions.

  1. AS, RT and RP are typically used for early prostate cancer patients. While some localized prostate cancer patients also require ADT, which is not considered in this study.
  2. The authors concluded that more patients after AS were treated with secondary treatment than RP and RT. But it appears that the treatment change ranged from 3.8% to 42.8%, which is too large to get a significant conclusion.

Other specific comments:

  1. Some diagrams could be required for the readers to better understand the conclusion.
  2. What is RCT in Line 184?

Author Response

Dear reviewer,

Thank you for taking the time to review our commentary and for allowing us to re-submit a revised version. Please find below replies to the comments as well as changes made to the manuscript.

We hope you find the changes satisfactory and that this article is now more suitable for publication.

Kind regards,

Katharina Beyer (on behalf of all co-authors)

---------------------------------------------------------------------------------------------------------------------------------------

Reviewer 1

By using PRIAS and ERSPC data, the authors try to compare the incidence of treatment changes in localized prostate cancer patients receiving different primary treatment, such as AS, RT or RP. Some issues were seen in the rationales and conclusions.

  1. AS, RT and RP are typically used for early prostate cancer patients. While some localized prostate cancer patients also require ADT, which is not considered in this study.

Re: In this study, we assessed the rate of biochemical recurrence (BCR) after treatment. BCR can follow after RT and RP, but ADT cannot. Therefore, we did not include ADT.

  1. The authors concluded that more patients after AS were treated with secondary treatment than RP and RT. But it appears that the treatment change ranged from 3.8% to 42.8%, which is too large to get a significant conclusion.

Re: As presented in section 3.2, the 3.8% refers to the cumulative probability of treatment change after 1 year, while the cumulative probability of treatment change is 42.8% after 5 years. Therefore, these numbers should be interpreted as cumulative probabilities (which are by definition increasing over time).

Other specific comments:

  1. Some diagrams could be required for the readers to better understand the conclusion.

Re: Cumulative incidence curves have been added in the supplementary material to increase the understanding of the conclusion.

  1. What is RCT in Line 184?

Re: Randomised Control Trial- we have amended this in the manuscript.

Reviewer 2 Report

I'm very happy to review the article entitled 'Secondary Treatment for Men with Localized Prostate Cancer: a pooled analysis of PRIAS and ERSPC-Rotterdam data within the PIONEER data platform'. The article is very well designed and written. 

The paper is original, reads well and is of importance to clinicians. The aim of the study and outcome measures are clearly defined with appropriate reference to the literature.

            The article aimed to get further insight into 76 the frequency of treatment switch after opting for AS and secondary treatments after 77 RP or RT in men diagnosed with early stage PCa. It’s a cohort study based on PRIAS and ERSPC-Rotterdam data platform was used. Primary outcome of this study was  threatment change following initial treatment. As expected  the authors found cumulative incidences of treatment change was higher in active surveillance than radical prostatectomy(RP) or raiotherapy(RT) in patients with PRostate Cancer. For RP and RT, change to secondary treatment was lower, but still possible with ten year incidences up to 20% and 16%.  The main strenght of this study larger number of the patients from two well-known databases were used to evaluate treatment change but heterogenity from two other populations is the main limitation. Despite this, as an authors stated the study one of the first studies comparing the incidence of guideline recommended treatment changes in men receiving different primary treatments (i.e., AS, RT, or RP) for localised PCa.

Author Response

Dear reviewer,

Thank you for taking the time to review our commentary and for allowing us to re-submit a revised version. Please find below replies to the comments as well as changes made to the manuscript.

We hope you find the changes satisfactory and that this article is now more suitable for publication.

Kind regards,

Katharina Beyer (on behalf of all co-authors)

----------------------------------------------------------------------------------------

Reviewer 2

I'm very happy to review the article entitled 'Secondary Treatment for Men with Localized Prostate Cancer: a pooled analysis of PRIAS and ERSPC-Rotterdam data within the PIONEER data platform'. The article is very well designed and written.

The paper is original, reads well and is of importance to clinicians. The aim of the study and outcome measures are clearly defined with appropriate reference to the literature.

            The article aimed to get further insight into 76 the frequency of treatment switch after opting for AS and secondary treatments after 77 RP or RT in men diagnosed with early stage PCa. It’s a cohort study based on PRIAS and ERSPC-Rotterdam data platform was used. Primary outcome of this study was  treatment change following initial treatment. As expected the authors found cumulative incidences of treatment change was higher in active surveillance than radical prostatectomy(RP) or radiotherapy (RT) in patients with PRostate Cancer. For RP and RT, change to secondary treatment was lower, but still possible with ten year incidences up to 20% and 16%.  The main strength of this study larger number of the patients from two well-known databases were used to evaluate treatment change but heterogenity from two other populations is the main limitation. Despite this, as an authors stated the study one of the first studies comparing the incidence of guideline recommended treatment changes in men receiving different primary treatments (i.e., AS, RT, or RP) for localised PCa.

Re: Thank you very much. We appreciate the feedback.

Reviewer 3 Report

To date, few studies evaluate the prevalence of need for secondary treatment in patients diagnosed with localized prostate cancer, over an extended period of time (10 years). Bringing together the end results of 2 reference trials, PRIAS and ERSPC, it adds great value to the reported data, allowing urologists to back-up active surveillance or radical treatment indications with sound data, making the patient an aware partner in the decision-making process.  

            Overall, the paper is well put together, with clear and adequate use of English language. The introduction presents concisely the importance of the discussed pathology and the current guidelines recommendations. For this section, I would like to suggest that the authors detail the pros and cons of each treatment option, by presenting them in a broader manner than already tackled in the first paragraph, rows 60 – 61.   

            Regarding Materials and methods, Study design, the authors chose to detail PRIAS and ERSPC inclusion criteria. I would suggest to present these trial information in the Introduction part, while focusing solely, for this section, on the inclusion and exclusion criteria for the current study. For the Outcomes part, the last sentence of the first paragraph, rows 117 – 118, would be a better fir for the Discussion segment. Lastly, Table 1 from Statistical analysis is more suitable for the Results section.

            The Results highlight the main significant findings in clear and concise manner. I think that this section can be enhanced by analyzing from a statistical standpoint the differences between active surveillance, radical prostatectomy and radiation therapy groups in terms of biochemical recurrence and incidence of treatment change. Additionally, some risk or prognosis factors might be derived from available data, that can be further correlated with recurrence or primary treatment chance.

Author Response

Dear reviewer,

Thank you for taking the time to review our commentary and for allowing us to re-submit a revised version. Please find below replies to the comments as well as changes made to the manuscript.

We hope you find the changes satisfactory and that this article is now more suitable for publication.

Kind regards,

Katharina Beyer (on behalf of all co-authors)

---------------------------------------------------------------------------------------------------------------------------------------

Reviewer 3

To date, few studies evaluate the prevalence of need for secondary treatment in patients diagnosed with localized prostate cancer, over an extended period of time (10 years). Bringing together the end results of 2 reference trials, PRIAS and ERSPC, it adds great value to the reported data, allowing urologists to back-up active surveillance or radical treatment indications with sound data, making the patient an aware partner in the decision-making process. 

            Overall, the paper is well put together, with clear and adequate use of English language. The introduction presents concisely the importance of the discussed pathology and the current guidelines recommendations. For this section, I would like to suggest that the authors detail the pros and cons of each treatment option, by presenting them in a broader manner than already tackled in the first paragraph, rows 60 – 61.  

Re: We have added additional detail on the treatments as suggested.

            Regarding Materials and methods, Study design, the authors chose to detail PRIAS and ERSPC inclusion criteria. I would suggest to present these trial information in the Introduction part, while focusing solely, for this section, on the inclusion and exclusion criteria for the current study. For the Outcomes part, the last sentence of the first paragraph, rows 117 – 118, would be a better fir for the Discussion segment. Lastly, Table 1 from Statistical analysis is more suitable for the Results section.

Re: We have amended the manuscript accordingly.

            The Results highlight the main significant findings in clear and concise manner. I think that this section can be enhanced by analyzing from a statistical standpoint the differences between active surveillance, radical prostatectomy and radiation therapy groups in terms of biochemical recurrence and incidence of treatment change. Additionally, some risk or prognosis factors might be derived from available data, that can be further correlated with recurrence or primary treatment chance.

Re: As presented in section 2.2, BCR was our primary outcome among men initially treated with RT/RP. Among men initially treated with AS, discontinuation due to protocol advise was our primary outcome. We did not decide to perform competing risk regression since our patient cohort consisted of men diagnosed with cT1-cT2, Gleason 3+3 and PSA ≤10.

Round 2

Reviewer 1 Report

The authors have addressed my concerns.